# Linking 16S rRNA Gene Classification to *amoA* Gene Taxonomy Reveals Environmental Distribution of Ammonia-Oxidizing Archaeal Clades in Peatland Soils

Haitao Wang,[a] Alexandre Bagnoud,[b] Rafael I. Ponce-Toledo,[b] Melina Kerou,[b] Micha Weil,[a] Christa Schleper,[b] Tim Urich[a]

aInstitute of Microbiology, University of Greifswald, Greifswald, Germany
bDepartment of Functional and Evolutionary Ecology, University of Vienna, Vienna, Austria

**ABSTRACT** A highly resolved taxonomy for ammonia-oxidizing archaea (AOA) based on the alpha subunit of ammonia monooxygenase (*amoA*) was recently established, which uncovered novel environmental patterns of AOA, challenging previous generalizations. However, many microbiome studies target the 16S rRNA gene as a marker; thus, the usage of this novel taxonomy is currently limited. Here, we exploited the phylogenetic congruence of archaeal *amoA* and 16S rRNA genes to link 16S rRNA gene classification to the novel *amoA* taxonomy. We screened publicly available archaeal genomes and contigs for the co-occurring *amoA* and 16S rRNA genes and constructed a 16S rRNA gene database with the corresponding *amoA* clade taxonomy. Phylogenetic trees of both marker genes confirmed congruence, enabling the identification of clades. We validated this approach with 16S rRNA gene amplicon data from peatland soils. We succeeded in linking 16S rRNA gene amplicon sequence variants belonging to the class *Nitrososphaeria* to seven different AOA (*amoA*) clades, including two of the most frequently detected clades (*Nitrososphaerales* $\gamma$ and $\delta$ clades) for which no pure culture is currently available. Water status significantly impacted the distribution of the AOA clades as well as the whole AOA community structure, which was correlated with pH, nitrate, and ammonium, consistent with previous clade predictions. Our study emphasizes the need to distinguish among AOA clades with distinct ecophysiologies and environmental preferences, for a better understanding of the ecology of the globally abundant AOA.

**IMPORTANCE** The recently established phylogeny of *amoA* provides a finer resolution than previous studies, allowing clustering of AOA beyond the order level and thus revealing novel clades. While the 16S rRNA gene is mostly appreciated in microbiome studies, this novel phylogeny is in limited use. Here, we provide an alternative path to identifying AOA with this novel and highly resolved *amoA* taxonomy by using 16S rRNA gene sequencing data. We constructed a 16S rRNA gene database with the associated *amoA* clade taxonomy based on their phylogenetic congruence. With this database, we were able to assign 16S rRNA gene amplicons from peatland soils to different AOA clades, with a level of resolution provided previously only by *amoA* phylogeny. As 16S rRNA gene amplicon sequencing is still widely employed in microbiome studies, our database may have a broad application for interpreting the ecology of globally abundant AOA.

**KEYWORDS** *amoA* gene taxonomy, phylogenetic congruence, AOA clades, peatland soils, 16S rRNA gene

The microbial transformation of ammonia to nitrite and nitrate, known as nitrification, plays a pivotal role in global nitrogen cycling (1). Ammonia oxidation, the first and rate-limiting step of nitrification, had been considered for a long time to be driven

Address correspondence to Haitao Wang, haitao.wang@uni-greifswald.de, or Tim Urich, tim.urich@uni-greifswald.de.

only by chemolithoautotrophic bacteria until the discovery and isolation of the first ammonia-oxidizing archaeon (AOA), *Nitrosopumilus maritimus* (2), as well as the identification of environmental sequences associated with archaea and related to the key enzyme ammonia monooxygenase (3, 4). Since then, AOA have been found to be ubiquitous and abundant in various environments using the marker gene *amoA* (ammonia monooxygenase subunit A), and cultivated representatives from both marine and terrestrial environments have been described (5–9).

Archaea carrying out ammonia oxidation belong to the taxonomically validated class *Nitrososphaeria* (10) and were initially assigned to a new phylum, *Thaumarchaeota* (11, 12), which was however recently disputed based on whole-genome phylogeny (13, 14). Within the class *Nitrososphaeria*, four orders (*Nitrosopumilales*, "*Candidatus* Nitrosotaleales," "*Candidatus* Nitrosocaldales," and *Nitrososphaerales*) have been described, and they are all represented by a small number of cultivated or enriched organisms (10) and a huge number of environmental sequences.

A recent study revealed that *amoA* has become the second most commonly sequenced marker gene after the 16S rRNA gene in studies of microbial ecology (15). Based on more than 30,000 *amoA* sequences, a highly resolved phylogeny with a novel multilevel taxonomy was established, which revealed novel environmental patterns of AOA that challenged previous generalizations (15). This phylogeny provides a finer resolution than previous studies, allowing clustering of AOA beyond the order level. The *Nitrososphaerales* $\delta$ clade (NS-$\delta$), for instance, by far the most frequently detected AOA clade, could not be clearly distinguished in former phylogenies because the diversity patterns of archaeal *amoA* genes were mostly inferred and interpreted based on customized divergent tree calculations (3, 16–21). Although some studies have started employing this newly developed reliable and robust *amoA* gene phylogeny (22–24), its usage and the resulting environmental patterns are still limited, since many microbiome studies target only the 16S rRNA gene as a marker. In order to get deeper insights into the diversity and environmental distribution of AOA clades, it would be imperative to link these two sequenced marker genes. The congruency between *amoA* and 16S rRNA gene phylogenies of AOA observed previously (15, 25) provides the framework for this linkage. To achieve this goal, constructing a database of genomes and contigs containing both genes is an essential starting point.

With only 3% coverage of the global land area, peatlands store 20 to 30% of the earth's soil organic carbon and nitrogen (26, 27). For decades, large areas of peatlands were drained for agricultural uses, resulting in a severe loss of stored carbon and nitrogen through the emission of greenhouse gases (28). The important greenhouse gas $N_2O$ is produced through microbial nitrification and denitrification (29, 30). The drainage and lowered water table in peat soils could lead to the increase of $N_2O$ emissions (26, 31), putatively due to the stimulated aerobic nitrification. Although rewetting was introduced recently to restore the carbon and nitrogen in the drained peatlands, it remains unclear how this process could influence the microbiomes and in particular bacteria and archaea involved in nitrogen cycling. By investigating microbial dynamics in three different peatlands in northern Germany, we recently found that rewetting significantly differentiated the drained and rewetted peat microbial communities, including both prokaryotes and eukaryotes (32). We also found that rewetting could potentially impact certain important microbial functions, including nitrification (32). Given that AOA mostly predominate among ammonia oxidizers in low-pH conditions (20, 33, 34), characterizing the AOA groups in acidic peatlands is important.

In this study, we aimed to create an AOA clade taxonomy for 16S rRNA genes and apply this to AOA in peatlands to better understand their ecological patterns. For this purpose, we first constructed a database based on the congruency between *amoA* and 16S rRNA genes by screening the publicly available genomic information of AOA, including metagenome-assembled genomes (MAGs) and contigs containing fragments of both genes. With this database, we used the 16S rRNA gene amplicon data set from our previous study (35) as an example to verify the accuracy of our methodology of

linking 16S rRNA gene to novel *amoA* clades identified in reference 15. Subsequently, we determined whether the environmental distributions of the 16S rRNA gene-defined clades in peat soils match the patterns observed in reference 15. Finally, the impact of environmental factors, including water status, peatland type, and soil depth, on these 16S rRNA gene-defined clades and their community composition were investigated under the hypothesis that the distributions of *amoA* clades are differentially affected by these factors.

## RESULTS AND DISCUSSION

**Database construction and linking 16S rRNA gene to *amoA* clades.** A 16S rRNA gene database (Text S1) was constructed based on the congruency between the 16S rRNA gene and *amoA* gene (Text S2) identified by screening published genomes (details are presented in Materials and Methods). Despite developing a comprehensive database, some AOA MAGs that might represent key members are missing in the database due to the lack of the 16S rRNA gene (see Materials and Methods), but this flaw will eventually disappear, as the database will be maintained with new genomes being sequenced. The current database comprises 124 representative 16S rRNA genes corresponding to four orders and their corresponding clades of the *amoA* gene. The four orders are "*Candidatus* Nitrosocaldales" (NC), *Nitrososphaerales* (NS), "*Candidatus* Nitrosotaleales" (NT), and (*Nitrosopumilales* (NP), while their corresponding clades are named with Greek letters following the order name (15). This database is the prerequisite for our first practical application of the congruency between 16S rRNA gene phylogeny and that of *amoA* since its first report (36), providing a framework for linking these two genes.

In our previous study, we found 49 amplicon sequencing variants (ASVs) belonging to *Nitrososphaeria* in peat soils (35) (Text S3). Using the constructed database, 44 ASVs could be assigned to a specific *amoA* clade (Table S1). In addition, we reconstructed maximum-likelihood (ML) phylogenetic trees of the 16S rRNA and *amoA* genes in our database in order to illustrate their congruency with respect to the delineation of *amoA* clades (Fig. 1), and confirm the *amoA* clade assignment of the 44 ASVs from our analysis (Fig. 1A). This also enabled us to identify and correctly place misannotated 16S rRNA gene sequences, as well as illustrating the uncertain position of symbiont-originating sequences due to their high evolutionary rates (Table S2).

By using the database, we found 7 of the *amoA* clades in the studied peatlands. The placement of the majority of the ASVs (i.e., 42 ASVs) on the phylogenetic tree confirmed their defined clade, illustrating the successful construction of the database. Interestingly, two of the ASVs initially assigned to NS-γ in our database formed a distinct clade within NS when placed in a phylogenetic tree. By including additional 16S rRNA gene sequences from environmental samples, we could verify that this placement was not an artifact due to the short length of the amplicon but rather represented an undefined/unidentified NS clade (NS-UD) (Fig. 1A). It is worth noting that not all clades defined by *amoA* gene phylogeny are represented in the 16S rRNA gene tree and database, as representative genomes or contigs are still lacking certain clades (e.g., NS-ε, NS-β, NP-β, and NP-ζ). It is therefore possible that this undefined/unidentified NS clade actually corresponds to *amoA*–NS-β or *amoA*–NS-ε. In total, we found 7 *amoA* clades in the studied peatlands in addition to one undefined clade (Fig. 1 and 2) and confirmed the predictive power of our database with sequences originating from relatively well-characterized *amoA* clades, while also illustrating the need to expand the genomic information on all AOA clades.

By linking *amoA* phylogeny to 16S rRNA sequences, we can now study AOA at a finer resolution, strongly contrasting previous AOA phylogenies, which resolved only order-level linkages and spare subclades, as illustrated by reference 15. For instance, NS-γ and NS-δ are the most abundant clades in soil environments, accounting for 13% and 23% of the total surveyed archaeal *amoA* sequences in public databases, respectively (Table 1); however, no enrichment or pure culture has been reported from these two clades, resulting in great interest in cultivating and thus understanding them.

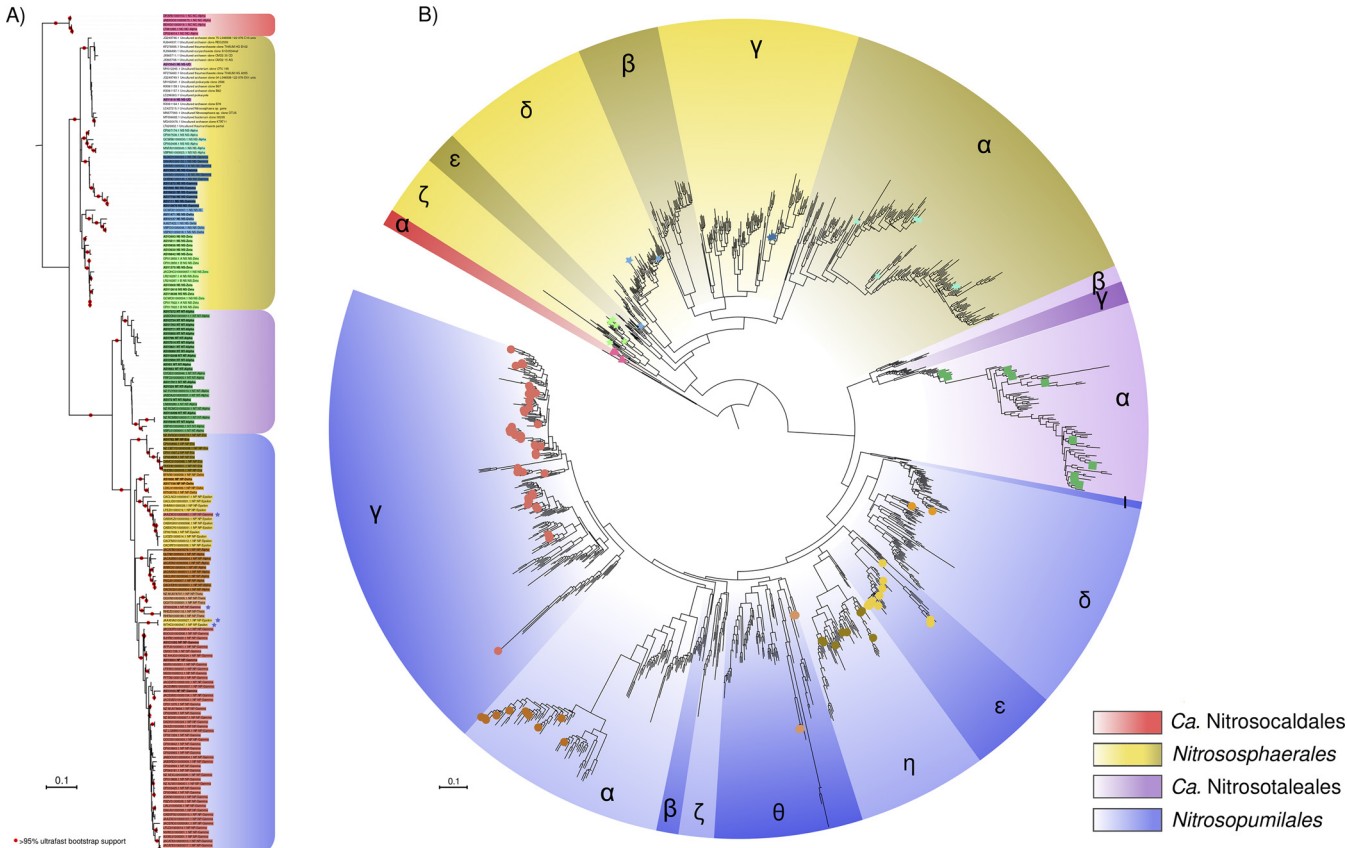

**FIG 1** (A) Phylogeny of 16S rRNA gene sequences. The maximum-likelihood phylogenetic tree was generated using IQTREE (v2.0-rc1) based on a curated alignment of ASV and 16S rRNA gene sequences comprising 1,434 nucleotide sites (see Materials and Methods). The names of the sequences or ASVs are colored with bars according to AOA clades, corresponding to the colors of the symbols in panel B. Stars depict sequences from host-associated AOA. The scale bar indicates the number of substitutions per nucleotide site. (B) Phylogeny of *amoA* sequences. The maximum-likelihood phylogenetic tree was reconstructed using an alignment of 594 nucleotide sites. 16S rRNA-*amoA* pair symbol codes are as follows: triangles, *amoA* sequences from "*Ca.* Nitrosocaldales"; stars, sequences from *Nitrososphaerales*; squares, sequences from "*Ca.* Nitrosotaleales"; circles, sequences from *Nitrosopumilales*. The colors of symbols are according to their 16S rRNA gene pair in panel A. Greek letters indicate the *amoA*-based annotation of AOA clades as in references 15 and 49. The scale bar indicates the number of substitutions per nucleotide site.

Given the lack of information on the 16S rRNA gene of these two clades, our method with the novel *amoA* phylogeny provides an alternative path to identifying these populations in appropriate environments.

With our database, we can define the *amoA* clades by 16S rRNA genes while accessing the information for other microbial groups using the same data set. One potential application could be utilizing the co-occurrence network to reveal the potential interactions between *amoA* clades and other microorganisms, thus better understanding the ecological roles of these clades in different environments.

**Distribution of the 16S rRNA gene-defined clades in studied peatlands.** Our study sites comprise three pairs of peatlands, namely, drained (AD) and rewetted (AW) alder forest, drained (CD) and rewetted (CW) coastal fen, and drained (PD) and rewetted (PW) percolation fen. Detailed site maps and descriptions can be found in references 32, 35, and 37. NT-$\alpha$ was the clade with the most extensive distribution in our data set, present in AD, AW, CD, CW, and PD (Fig. 2). It was also the most abundant clade in these sites, with relative abundance being either stable (AD and AW) or increasing with depth (CD, CW, and PD) (Fig. 2). The pH in these sites was below 6 (Fig. S1), in line with the observation that most NT-$\alpha$ members prefer acidic conditions (pH < 6.5).

The two most often sequenced terrestrial clades, NS-$\gamma$ and NS-$\delta$, were present in our peatland soils (Fig. 2), supporting their prevalence in soils. The presence and high relative abundance of NS-$\gamma$ in the alder forest suggest that peatlands in alder forests might be good sites for studying this undercharacterized clade, or even nitrifiers in general,

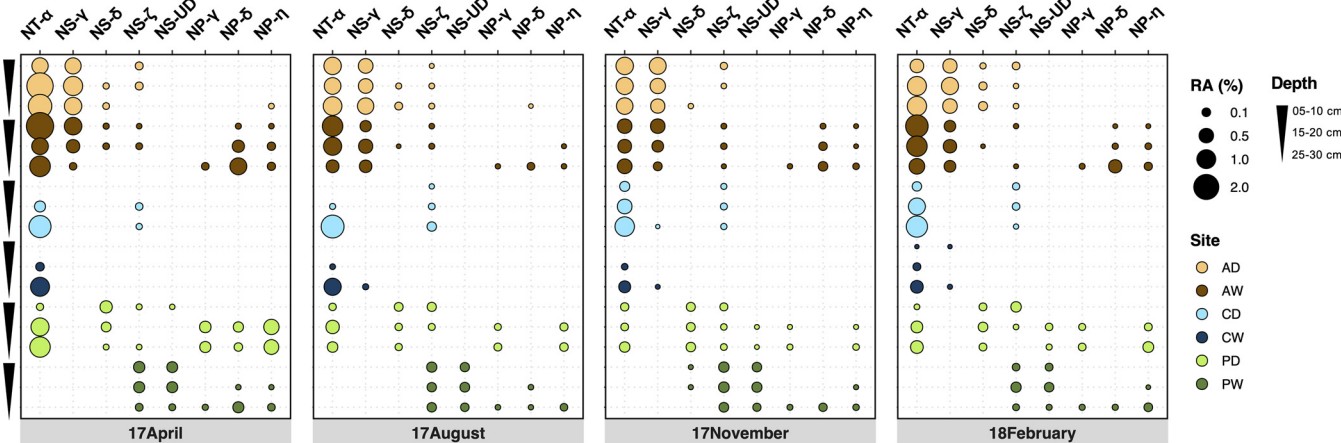

**FIG 2** Relative abundance of 16S rRNA gene-defined clades to the total prokaryotes. The bubble plots show the changes of relative abundance in different seasons, depths, and sites. RA, relative abundance.

since this site had the highest relative abundances of both AOA and ammonia-oxidizing bacteria (AOB) (Fig. S2 and S3). NS-$\gamma$ was also reported to occur in acidic conditions, matching the pH of ~6 occurring in the alder forest sites (Table 1). Although little is known about NS-$\delta$, globally the most detected clade in soils, they seem to generally prefer neutral pH (15), which might explain their low abundance in our acidic peatland soils (Fig. 2). The most recent enrichment culture of NS-$\zeta$ from an arctic fen peatland showed optimum growth at pH 6 to 7 (24), supporting the occurrence of this clade in our percolation fen with a similar pH range (Fig. S1). NS-UD was found only in the percolation fen, with a higher abundance in the rewetted site (Fig. 2). This undefined clade could potentially be NS-$\beta$ or NS-$\varepsilon$, both of which are still poorly characterized, as the pH and habitat preferences of these two clades (15) also match the profile of PD and PW (Fig. S1). Their presence also suggests the percolation fen as a potential resource for studying them.

As illustrated by reference 15, NS and NT are predominantly found in freshwater or soils and sediments (Table 1). Although NP is encountered mainly in marine environments, some species/clades were also found in freshwater or soils and sediment (Table 1). For instance, NP-$\eta$, which is also present in the four different peatlands, was primarily found in freshwater systems (15). Interestingly, none of the NP clades was found in the coastal fen sites, which are characterized by higher salinity. One reason might be the more acidic conditions in the coastal fen sites as well as in PD, where few NP clades were found (Fig. 2 and Fig. S1). Typically, the pH in the ocean is more neutral

**TABLE 1** Summary of the distribution patterns of the seven archaeal *amoA* clades observed in reference 15 that were found in peat soils (this study)

| Clade[a] | Proportion (%)[b] | Cultivability[c] | Habitat[d] | pH[e] |
|---|---|---|---|---|
| NT-$\alpha$ | 6 | ++ | SF | Mostly acidic |
| NS-$\gamma$ | 13 | − | SF | Mostly acidic |
| NS-$\delta$ | 23 | − | SF | Mostly neutral |
| NS-$\zeta$ | 1 | ++ | SF | Acidic/alkaline |
| NP-$\gamma$ | 15 | ++ | MC/SF | |
| NP-$\delta$ | 5 | − | MC/SF | |
| NP-$\eta$ | 8 | + | F | |

[a]NP, *Nitrosopumilales*; NS, *Nitrososphaerales*; NT, "*Ca.* Nitrosotaleales."
[b]Ratio of a certain clade to all observed AOA sequences found in reference 15.
[c]Availability of a pure or enrichment culture in a clade. −, no (enrichment) culture; +, only enrichment culture; ++, pure culture.
[d]Environments where most *amoA* genes from a clade were found. MC, marine and coastal; S, soils and sediments; F, freshwater.
[e]The pH condition in which most of the AOA from a clade were found or cultivated. The pH conditions acidic (pH < 6.5), neutral (6.5 ≤ pH ≤ 7.5), and alkaline (pH > 7.5) are defined in reference 15.

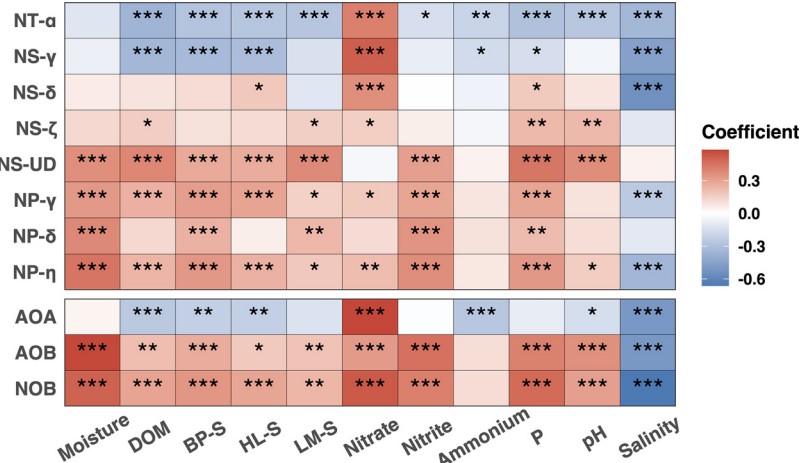

**FIG 3** Heat map showing the Spearman's rank correlations between different soil parameters and relative abundance of nitrifiers and 16S rRNA gene-defined clades. Asterisks indicate the significance of the correlations (*, $P < 0.05$; **, $P < 0.01$; ***, $P < 0.001$). DOM, dissolved organic matter; BP-S, biopolymer substances; HL-S, humic-like substances; LM-S, low-molecular-weight substances.

to slightly alkaline, which marine clades might be more adapted to. Moreover, the site with the highest salinity (CW) had the lowest redox potential, indicating a more anoxic condition that could inhibit AOA in general.

Overall, the distribution patterns of these 7 *amoA* clades in the studied peatlands matched the observed patterns in the previous study (15) in the context of habitat specificity and soil pH.

**Correlations between physicochemical factors and the *amoA* clades.** The distribution pattern of these *amoA* clades showed few differences among the four seasons, while the pattern within each season was quite different across sites and depths (Fig. 2). The alder forest was mainly dominated by the two most abundant clades in our study, NT-$\alpha$ and NS-$\gamma$, while all clades occurred in the percolation fen, with a preference in either PD or PW (Fig. 2 and Fig. S2).

Previous studies have shown that NS-$\gamma$ appears to respond positively to $NH_4^+$, shown as a large increase of potential gross nitrification rates in "clade B"-dominated soils with the addition of $NH_4^+$ (38). This clade was also found to dominate nitrogen-fertilized soils (39). AD and AW showed the highest nitrate and ammonium concentrations, respectively (Fig. S1). The drier condition might benefit the aerobic nitrification in AD, resulting in a higher nitrification rate and thus transformation of ammonium to nitrate, promoting the production of nitrate in AD. These concentrations together indicate a high background level of ammonium in the alder forest, which is potentially responsible for the larger amount of NS-$\gamma$ (Fig. 2).

High salinity showed selection for the AOA community, as shown by the fact that few clades other than NT-$\alpha$ were present in the coastal fen (Fig. 2). The presence of NT-$\alpha$ indicated their adaption to saline conditions, supported by a previous study showing that the abundance of *Nitrosotalea*-like operational taxonomic units (OTUs) increased with increasing salinity (40). While adaptations to high salinity are encountered in many lineages within *Nitrosopumilales* (41), intriguingly, no NP representatives were found in these sites, indicating additional inhibitory conditions, as discussed above.

Water status showed a significant impact on AOA species belonging to these different *amoA* clades (Fig. S2). For instance, the NT-$\alpha$ ASVs were overrepresented in both AD and AW, while they were more important in the drained sites of the coastal fen and percolation fen than in their rewetted counterparts. The lower pH in the drained sites (Fig. S1) might contribute to the overrepresentation of the NT-$\alpha$ ASVs in drained sites, since organisms in this clade favor acidic conditions (15). This is further supported by the negative correlation between the abundance of this clade and soil pH (Fig. 3). The

NS-$\gamma$ ASVs were more important in AD than in AW. The two NS-$\delta$ ASVs were more important in PD than in PW, while NS-$\zeta$ ASVs could be overrepresented in both PD and PW and in CD. Only two NP ASVs were found to be overrepresented in AW. These divergent responses to rewetting suggest different preferences of these clades with regard to water status, which might be due to differing soil conditions resulting from rewetting.

An interesting pattern emerges when the distribution of NS-$\zeta$, which is present almost exclusively in PW and the upper layers of PD, is examined more closely. These sites have the highest content of dissolved total organic matter (DOM), including biopolymer substances (BP-S), humic like substances (HL-S) and low-molecular-weight substances (LM-S), and P (Fig. S1) from all sites, and this clade is positively correlated with all except HL-S (Fig. 3). Genomic analyses have shown that this clade might harbor alternative pathways for energy generation, as representatives encode pyrroloquinoline quinone (PQQ)-dependent glycose/sorbosone family dehydrogenases responsible for the oxidation of sugars or other organic acids (24), among other indications. In addition, NS in general and NS-$\zeta$ in particular are enriched in F420-dependent luciferase-like hydride transferases (LLHT), enzymes especially capable of degrading recalcitrant organic matter and humic-like substances (41–43), as well as in glyoxalase/bleomycin resistance/dioxygenase superfamily proteins involved in xenobiotic degradation and detoxification (41). Additionally, they contain multiple rRNA operons, suggesting resilience to environmental changes. These observations are supported by the habitat preference of isolates of this clade, namely, contaminated soils high in organics, wastewater treatment plants (WWTP), and fertilized agricultural soils (8, 44, 45). The negative and positive correlation of HL-S with NS-$\gamma$ and NS-$\delta$, respectively (Fig. 3) could offer hints for the genomic repertoire of these two clades with respect to their detoxification capacities.

NP clades showed contrasting correlation trends compared to the major clades, NS-$\gamma$ and NT-$\alpha$, as all three NP clades correlated positively with DOM constituent compounds such as BP-S and LM-S, while NP-$\gamma$ and NP-$\eta$ correlated additionally with HL-S and total DOM (Fig. 3). NP-$\gamma$ is the most cosmopolitan NP clade, encountered in all moderate environments; NP-$\eta$ is found mostly in terrestrial/freshwater habitats, with characterized representatives isolated from WWTP, soils, and thermal environments, while the understudied NP-$\delta$ is predominantly found in estuarine/coastal and marine sediments (15). While all cultured representatives of these clades were shown to be autotrophs, recent investigations into NP-$\delta$ and NP-$\gamma$ draft genomes from deep marine sediments and the deep ocean revealed mechanisms of supplementation of the central carbon metabolism with organics (such as amino acids and fermentation products), raising the possibility of identifying novel mechanisms of tolerance and usage of DOM in NP lineages (46–49). Their occurrence in the studied peatlands might also provide novel insights in understanding these mechanisms.

Interestingly, the correlation profile of the clade NS-UD was more similar to the profiles of NP clades than of the NS clades. The highly significant positive correlations ($P < 0.001$) to DOM, BP-S, HL-S, and LM-S might indicate that this understudied clade also encodes capabilities of organic carbon usage, as in the aforementioned NP clades. NS-UD, as well as the other NP clades, shows significant positive correlations to phosphorus (Fig. 3), while PD and PW contain the highest phosphorus levels of all sites, with concentrations similar to the ones encountered in P-enriched regions in the ocean (50). Analyses of the distribution of NP clades in the ocean, where phosphorus concentrations vary with depth, revealed that P availability is a key factor shaping the genomic repertoire of marine AOA and influencing their distribution (50). Perhaps a similar situation is encountered here, where the locally higher availability of P influences the dominant AOA populations. Finally, NS-UD and NP-$\delta$ are the only clades that do not show a significantly negative correlation to salinity, indicating the possible existence of adaptations to high osmolarity. In summary, the profile of this unidentified

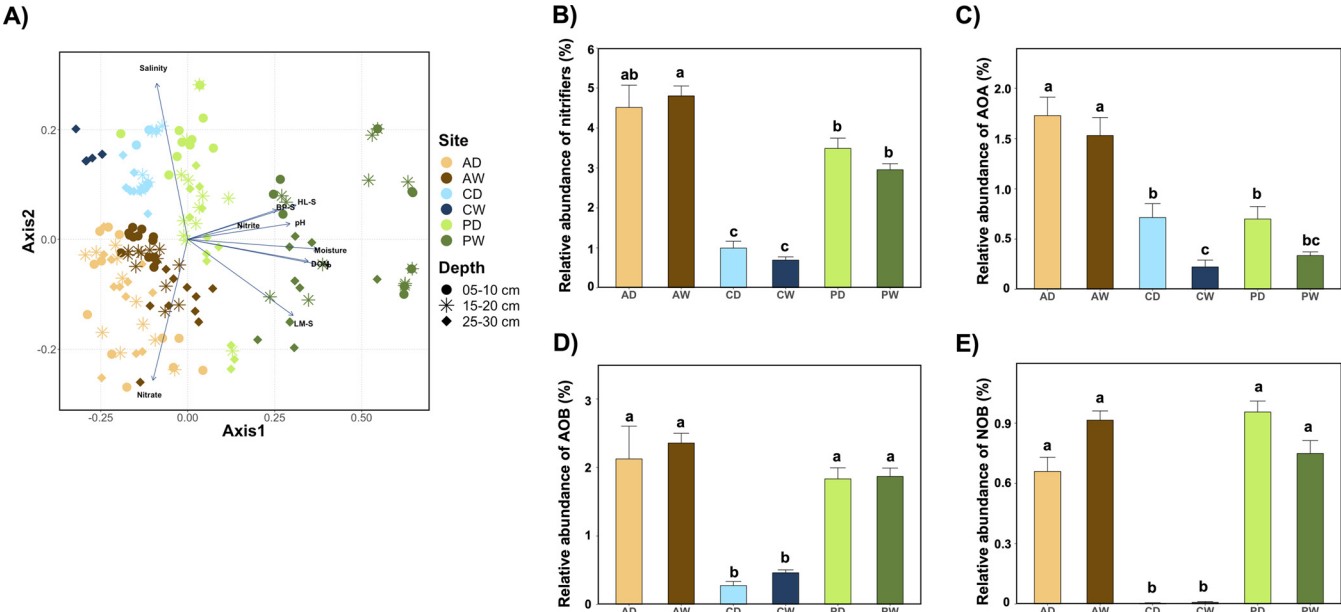

**FIG 4** (A) NMDS plot showing the community structure of AOA. The main ordinations show similarity between samples, and the arrows show correlations between environmental variables and ordination axes. DOM, dissolved organic matter; BP-S, biopolymer substances; HL-S, humic-like substances; LM-S, low-molecular-weight substances. (B to E) Relative abundances of total nitrifiers (B), AOA (C), AOB (D), and NOB (E) in different sites. Data are means, and error bars indicate standard errors. Different letters indicate significant differences (Kruskal-Wallis *post hoc* test; $P < 0.05$).

clade exhibits interesting similarities with those of the NP clades, raising questions as to its ecophysiological potential.

The two major clades NT-$\alpha$ and NS-$\gamma$ showed positive correlations with nitrate but negative correlations with the other soil properties, in contrast to the other clades (Fig. 3). However, the overall AOA abundance showed a similar pattern driven by these two major clades (Fig. 3). The positive correlations between these two major clades and nitrate, and the negative correlations between them and ammonium and pH, supported physiological preferences of these clades, as observed before (15, 38, 51). The contrasting correlations of the different AOA clades emphasizes different ecophysiologies and the need to distinguish carefully between clades instead of referring to all AOA as one group in correlation studies.

**Temporal and spatial variations of total nitrifying communities.** The nonmetric multidimensional scaling (NMDS) plot shows that the community composition of AOA was distinct between the three peatland types as well as between drained and rewetted sites (Fig. 4A). In PD and AW, there was also an obvious transect from upper to deeper layer (Fig. 4A). Permutational multivariate analysis of variance (PERMANOVA) suggested that peatland type was the biggest driver in changing the community composition ($R^2 = 0.276$; $P = 0.001$), followed by rewetting ($R^2 = 0.077$; $P = 0.001$) and depth ($R^2 = 0.044$; $P = 0.001$). Season showed no significant impact on the community composition ($R^2 = 0.015$; $P = 0.168$). The community composition was also driven by soil properties. Apparently, salinity and nitrate were more associated with the coastal fen and the alder forest, respectively, while the other soil properties were more related to the percolation fen (Fig. 4A). These factors have also been found to be significant drivers of AOA community composition in other studies (38, 52–55), especially salinity, which is a key factor driving the nitrogen cycling in coastal sediments (17). Therefore, these soil factors mediated the effect of rewetting and peatland type on the community composition of AOA, and also that of the whole prokaryotic and eukaryotic communities (32).

The relative abundance of AOA declined during summer and autumn compared with spring and winter, although these changes showed limited significance (Fig. S3). Our previous study showed a contrasting pattern of changes in DOM between these

seasons (35), resulting in negative correlations between abundances of AOA and DOM (Fig. 3). The higher decomposition rate in summer and autumn resulting from higher temperatures may lead to higher concentrations of organic carbon, which could further inhibit nitrifying activities (56–58). Similarly, the higher DOM concentration in the percolation fen (Fig. S1) could also have contributed to the lower abundance of nitrifiers (AOA) in this fen compared to the alder forest (Fig. 4B and C). NS-$\gamma$ and NT-$\alpha$ drove the overall pattern of AOA and were responsible for DOM impact. While the coastal fen showed less organic carbon compared to percolation fen (Fig. S1), nitrifier abundance was the lowest in the former (Fig. 4B to E). This might be due to the highest salinity in the coastal fen (Fig. S1), which could also inhibit nitrifying activities.

Interestingly, the relative abundance of AOA was generally higher in the drained sites than in the rewetted sites, with a significant difference between CD and CW (Fig. 4C). To our knowledge, few studies have reported the impact of long-term rewetting on nitrifying communities. While our study focused on decades of drainage and then rewetting, our results still support the findings that nitrifying activity or communities could be stimulated during the drought periods in dry-wet cycles (59–61). Moreover, the insignificant changes of AOB abundance between drained and rewetted sites (Fig. 4D) support the idea that AOA are less resistant and resilient to drying-rewetting than AOB in soils (62).

We previously observed that the water levels in AW and PW were higher than those in all the other sites (32). Interestingly, the abundance of nitrifiers decreased with increasing depth in these two rewetted sites, while it increased with increasing depth in the other sites (Fig. S4). In AW and PW, the upper layers of the soil might still receive oxygen diffusing from the atmosphere through the water layer, while the waterlogged sublayers of the soil might be oxygen deficient, resulting in the lower abundance of aerobic nitrifiers in deep layers. However, oxygen might still diffuse to sublayers due to the lower water table in other sites. Previous studies have also observed increased AOA abundances with depth, identifying this environmental factor, and the conditions represented by it (energy, water content, redox state, etc.), salinity, and total N as major determinants of AOA distribution (55).

**Conclusions.** This is the first study to provide a robust database for exploiting the phylogenetic congruence of archaeal 16S rRNA and *amoA* genes. With this framework, we were able to link 16S rRNA gene diversity to distinct *amoA* clades with a comparative phylogeny. The distributions of the 16S rRNA gene-defined clades in the studied peatland soils matched the patterns revealed by a previous study (15) with regard to habitat specificity and soil pH. On the ecosystem level, our analyses showed that rewetting the drained peatlands had a significant impact on the distribution of these *amoA* clades as well as the whole AOA community structure consisting of these clades, mainly driven by pH and concentrations of nitrate and ammonium. Our study provides novel insights for interpreting the ecology of globally abundant AOA, using a 16S rRNA gene sequencing data to access the level of resolution provided previously only by *amoA* phylogeny and emphasizes the need to distinguish different AOA clades given their different and distinct correlations with environmental factors.

## MATERIALS AND METHODS

**Construction of the database.** We extracted archaeal genomes containing both the 16S rRNA gene and *amoA* gene from the databases, allowing the assignment of 16S rRNA sequences to corresponding *amoA* clades. We screened published genomes from NCBI, including 333,502 archaeal nucleotides (length $\geq$ 2,000 bp), 1,106 archaeal RefSeq genomes, and 5,736 archaeal GenBank genomes (including MAGs). We found that some MAGs (e.g., from reference 63) found in the initial screening lacked the 16S rRNA gene, and they were therefore excluded. In total, 135 linked pairs of 16S rRNA and *amoA* genes could be identified. The database was constructed by extracting the 16S rRNA gene sequences from these identified genomes or long sequences. After phylogenic analysis of 16S rRNA genes (see below), some sequences that were identified as contaminants or symbiont-originating sequences were removed (Table S2). Due to a limited number of the published genomes, only two 16S–NP-$\delta$ pairs were identified. However, we required at least three reference sequences for each clade while assigning using UCLUST. We further included one more sequence of this clade by using its 16S rRNA gene sequence as the query in online BLASTN searches to retrieve one the most closely related environmental sequence. Finally, a database comprising 124 16S rRNA gene sequences (Text S1) corresponding to 124 linked pairs of 16S rRNA and *amoA* genes (Text S2) was established. The corresponding *amoA* clade names of each 16S

rRNA gene sequence were annotated according to reference 15 based on the paired *amoA* sequences (Text S2). The pipeline for the database construction, which can also be used for updating the existing one, is available on GitHub (https://github.com/alex-bagnoud/Linking-16S-to-amoA-taxonomy/).

**Study sites, sampling, and sample processing.** Our study sites comprise three pairs of drained (D) and rewetted (W) peatlands, including alder forest (AD and AW), coastal fen (CD and CW), and percolation fen (PD and PW), which represent typical landscapes in northern Germany. Detailed information for these sites, including sampling and sample processing methods, was provided in previous studies (32, 35, 37). Briefly, soil samples were taken during four seasons, including April, August, and November of 2017 and February of 2018, at three depths (5 to 10 cm,15 to 20 cm, and 25 to 30 cm). The corresponding soil edaphic properties, including moisture, pH, salinity, dissolved organic matter (DOM), and phosphorus, $N-NH_4^+$, and $N-NO_x^-$ concentrations, were measured. Soil DNA was extracted, and the 16S rRNA gene was amplified with the EMP (Earth Microbiome Project) primer pair 515F/806R (64), targeting both archaea and bacteria. Amplicons were sequenced with an Illumina MiSeq 300-bp paired-end platform. The 16S rRNA amplicon sequences were processed with the dada2 (v1.8.0) pipeline (65) in R v3.5. Finally, 25,864 amplicon sequence variants (ASVs) were obtained from 209 samples (details in reference 35). Each ASV was taxonomically assigned against a modified version of the SILVA SSUref_NR_128 database using CREST (66).

**Linking 16S rRNA to *amoA* gene taxonomy.** Forty-nine ASVs belonging to the class *Nitrososphaeria* were used for *amoA* clade assignment. The representative sequences of these ASVs (Text S3) were taxonomically assigned against the 16S rRNA gene database constructed as aforementioned using QIIME (67). The UCLUST consensus taxonomy assigner was used. Sequence with a similarity to the database sequences above 90% was considered a hit. The taxonomic assignment took place only when over 51% of the hits ($\geq$3) of an input sequence matched a specific assignment. The detailed command can be found on GitHub (https://github.com/alex-bagnoud/Linking-16S-to-amoA-taxonomy/).

**Phylogenetic analysis of 16S rRNA gene sequences.** The ASV3343 and ASV1618 sequences belonging to an undefined *Nitrososphaerales* clade for which we do not have representatives in our 16S rRNA gene database were used as queries in online BLASTN searches to retrieve closely related environmental sequences. The 10 top best hits of each BLAST run were selected and added to the 16S data set together with the ASV reported in this study. The 16S rRNA gene sequences were aligned using the SINA aligner (SINA v1.2.11) of the SILVA online server (68) followed by manual trimming of the alignment in AliView (69). The cleaned alignment was used to reconstruct a maximum-likelihood tree in IQTREE (v2.0-rc1) (70) with 1,000 ultrafast bootstrap replicates and the SH-like approximate likelihood ratio test ("-alrt 1000") (71). The best-fit model (TIM3e+R3) was selected by ModelFinder ("-m MFP") (72) according to the Bayesian information criterion (BIC).

**Phylogenetic analysis of *amoA* sequences.** A total of 123 *amoA* nucleotide sequences (Text S4) were added to the reference alignment of *amoA* reported in reference 15 using MAFFT v7.427 ("–add" parameter) (73). IQTREE (v2.0-rc1) (70) was used to reconstruct the maximum-likelihood phylogenetic tree of *amoA* sequences with the GTR+F+I+G model of sequence evolution under a constrained tree search ("-g" parameter) based on the phylogenetic tree reported in reference 15 with 1,000 ultrafast bootstrap replicates and the SH-like approximate-likelihood ratio test ("-alrt 1000") (71).

**Identification of AOB and NOB.** Assigning taxonomy against the Silva database resulted in too-broad groups of AOB and nitrite-oxidizing bacteria (NOB) whose sequences were not all identified as being from characterized AOB or NOB. Therefore, AOB and NOB were further identified using BLASTN with the 16S rRNA gene database from NCBI according to a previous study (74). ASVs identified as AOB were at least 93% identical to known nitrifying species within the family *Nitrosomonadaceae* (*Betaproteobacteria*) and genus *Nitrosococcus* (*Gammaproteobacteria*). ASVs identified as NOB were at least 90% identical to known species within the phylum *Nitrospirae*.

**Statistical analysis.** The statistical analyses were performed with R v3.5. A nonparametric Kruskal-Wallis test was used to test the significance of the difference among different seasons or depths with the vegan package. Kruskal-Wallis *post hoc* tests were performed to compare the means of soil properties between each two sites using the PMCMR package. The ternary plot was used to show the different relative abundances of each AOA ASV in three different peatland types with the ggtern package. The AOA ASVs that were differentially abundant in drained and rewetted sites were identified with differential expression analysis based on the binomial distribution using the DESeq2 package. The community composition was analyzed using nonmetric multidimensional scaling (NMDS) based on Bray-Curtis dissimilarity distances after normalizing AOA ASV counts with the Hellinger transformation. Permutational multivariate analysis of variance (PERMANOVA) was conducted to test the significance of impacts of rewetting, peatland type, depth, and season on the AOA community composition using the vegan package. Spearman's rank correlations were conducted to examine the relationships between abundances of nitrifiers and soil parameters using the vegan package and were plotted with the pheatmap package. The soil parameters were fitted into NMDS ordination, and the significant ones ($P < 0.05$) were kept using the envfit function in the vegan package. All the *P* values for multiple comparisons were adjusted by the false-discovery-rate (FDR) method, and the null hypothesis was rejected when *P* values were less than 0.05.

**Data availability.** All data generated or analyzed during this study are included in this article and the supplemental material. The detailed script for constructing the database and running the assignment analysis can be found on GitHub (https://github.com/alex-bagnoud/Linking-16S-to-amoA-taxonomy/).

## SUPPLEMENTAL MATERIAL

Supplemental material is available online only.

**TEXT S1**, TXT file, 0.2 MB.

**TEXT S2**, TXT file, 0.01 MB.

**TEXT S3**, TXT file, 0.01 MB.
**TEXT S4**, TXT file, 0.1 MB.
**FIG S1**, PDF file, 0.2 MB.
**FIG S2**, PDF file, 0.2 MB.
**FIG S3**, PDF file, 0.4 MB.
**FIG S4**, PDF file, 0.4 MB.
**TABLE S1**, XLSX file, 0.01 MB.
**TABLE S2**, XLSX file, 0.01 MB.

## ACKNOWLEDGMENTS

This study was supported by the European Social Fund (ESF) and the Ministry of Education, Science and Culture of Mecklenburg-Western Pomerania (Germany) within the scope of the project WETSCAPES (ESF/14-BM-A55-0034/16 and ESF/14-BM-A55-0030/16). T.U. and H.W. acknowledge additional support by the DFG (German Research Foundation, project number TU/4-1).

C.S. and T.U. conceived and designed the research; H.W. and M.W. collected the samples and performed the lab work; A.B. developed the pipeline and constructed the database; R.I.P.-T. constructed the phylogenetic trees; M.K. helped to generate the structure of the manuscript and commented on ecophysiologies of AOA clades; H.W. analyzed the data and wrote the first draft of the manuscript, and all authors contributed to revisions. All authors read and approved the final manuscript.

We declare that we have no competing interests.

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
