## [Reviewer comments · mSystems]

Linking 16S rRNA gene classification to amoA gene taxonomy reveals environmental distribution of ammonia-oxidizing archaea clades in peatland soils

Haitao Wang, Alexandre Bagnoud, Rafael Ponce-Toledo, Melina Kerou, Micha Weil, Christa Schleper, and Tim Urich

Corresponding Author(s): Haitao Wang, University of Greifswald

Review Timeline:

Submission Date:	May 1, 2021
Editorial Decision:	June 23, 2021
Revision Received:	July 8, 2021
Editorial Decision:	August 5, 2021
Revision Received:	August 10, 2021
Accepted:	August 12, 2021

Editor: Sean Gibbons

Reviewer(s): Disclosure of reviewer identity is with reference to reviewer comments included in decision letter(s). The following individuals involved in review of your submission have agreed to reveal their identity: Jennifer Rocca (Reviewer #1)

Transaction Report:

DOI: <https://doi.org/10.1128/mSystems.00546-21>

June 23, 2021

Dr. Haitao Wang
University of Greifswald
Greifswald
Germany

Re: mSystems00546-21 (Linking 16S rRNA gene classification to amoA gene taxonomy reveals environmental distribution of ammonia-oxidizing archaea clades in peatland soils)

Dear Dr. Haitao Wang:

Thank you for submitting your manuscript to mSystems. We have completed our review and I am pleased to inform you that, in principle, we expect to accept it for publication in mSystems. However, acceptance will not be final until you have adequately addressed the reviewer comments. I initially invited three reviewers, but the second reviewer did not provide their review in a timely fashion, so I'm making my decision based upon comments from the first and third reviewers. While both reviewers thought the manuscript was relatively well-written, they identified several grammatical and language errors throughout the text. Please ensure that you carefully revise the text to improve the quality of the writing. Another concern brought up by the third reviewer was the usefulness of the network analysis. Similar to the reviewer, I think this section should be removed from the manuscript, as it doesn't appear to add to the paper in a substantive way. I suggest you either delete that section entirely or move it to the supplement.

Preparing Revision Guidelines

For complete guidelines on revision requirements, please see the Instructions to Authors at <https://msystems.asm.org/sites/default/files/additional-assets/mSys-ITA.pdf>. **Submissions of a paper that does not conform to mSystems guidelines will delay acceptance of your**

manuscript.

Sincerely,

Sean Gibbons

Editor, mSystems

Journals Department
Reviewer comments:

Reviewer #1 (Comments for the Author):

In this manuscript, the authors present an innovative approach to link the curated taxonomy of the archaeal ammonia monooxygenase gene with each corresponding 16S rRNA ASVs. Then, they used these connections to infer shifts in AOA subclade abundance with changes in environmental conditions across several peatland soil in N. Germany. Using network analysis, they also tracked changes in bacterial community structure with correlated abundance patterns to the amoA set of ASVs, and used trait information for these bacterial clades to propose putative reasons for their correlated abundance patterns. This is an interesting piece and enables the assessment of the degree of niche specialization of AOA here applied to peatlands, but it will be exciting to see how these patterns hold in other environments.

The development of the database linking amoA genes to their 16S rRNA counterparts is rigorous, and the subsequent application to understanding AOA community shifts is comprehensive. Aside from several grammatical errors (see non-exhausted list below), one minor concern or consideration to potentially address is that while this database certainly further delineates AOA into finer subclades, it may still lack some key members by virtue of the fact that these amoA-16S MAGs (or long contigs) are missing (lines 422-423).

Assuming that MAGs are more often formed from more abundant clades, the database is missing rarer members. I guess this isn't problematic as the objective here is to understand function-community relationships, and these members are less critical to those patterns. However, another set of missing MAGs (or long contigs) are those for which the amoA and 16S genes are very

distantly spaced, or have high homopolymer stretches between them? I am naive to the conservation of the basepairs between these two genes, but are there ITS-like regions? If so, it is conceivable that AOAs could still be missing from the database because amoA-16S MAGs don't exist yet. Again, this may not be a reasonable concern, but if it is, maybe it is worth mentioning somewhere that despite developing a comprehensive database, some key members may still be missing.

Line-specific comments:

457 - 90% seems like a low threshold, but maybe it is sufficient?

506-508 - I assume the Hellinger-transform data was used as input to both Deseq and the ordination, not just the latter?

527 - "Containmination"

Fig 3 - "Corefficient" (I actually thought this was a new term)

Reviewer #3 (Comments for the Author):

This is a very welcome study which aims to provide unity between amoA and 16S rRNA-gene based phylogenies in molecular ecology surveys of AOA. While the phylogenies produced from these two genes have long been recognized as highly congruent, the inability to directly use, for example, the Alves et al. classification system with 16S rRNA data beyond order-level assignment has limited the interpretation. The demonstration of the using this 16S rRNA-based approach for looking at the distribution of clades in the different samples across time and depth (Figure 2) was really very nice and clear and shows that this will be a welcome tool for the community. While the manuscript is generally clear and logically structured, I do have some minor comments.

1. The term '16S-predicted amoA clades' is used in several places in the manuscript. While I understand what it means (i.e. clades of AOA that were named from an amoA-based phylogeny but are now identified from their paired/corresponding 16S rRNA sequences) it is a bit confusing and doesn't quite make sense. At this point in the manuscript it is pretty clear (to me) that these the 16S rRNA gene-defined clades correspond to the same groups (names) as the original amoA-based designation. Can you just say 16S rRNA gene-defined clade?

2. The study makes heavy use of correlation to make predictions of which physicochemical properties influence the individual clades. I think this is okay (although terms like "Associations" (title in line 218) are a little strong for me) if being used for making hypotheses for experimentally testing in the future. However, I'm less sure the network analysis is as compelling or informative as the rest of the manuscript, with points in the discussion understandably vague. These network correlations are not proof (or necessarily an indication) of interaction, and statements that correlating Acidobacteria and Nitrosotalea populations are both slow-growing and acidophilic doesn't really tell us anything (in my opinion). I would consider simply removing the network component, as the (co-?)-correlations seem to be with many different taxa and not really indicating any causal or functional interactions.

3. The manuscript is well written although some grammatical/structural issues occurred in the last sections of the discussion and in the M&M. For the latter I have highlighted suggested revisions for the first paragraph although these are not exhaustive.

4. Figure 1A - what do the coloured sequence names imply? For example, in the Nitrosotaleales, the sequence names are in green. I guess this is a habitat colour-coding, but details are missing in the

key or legend.

Minor comments

34 - as 'a' marker

43, 238 etc. - Water content or status rather than condition?

67 - The microbial transformation of ammonia..... ?

69 - had been

83 - suggest removing 'right' - redundant

133 - suggest '16S rRNA-predicted'

196 - I realise pH 6 is technically acidic, but in the context of soils, are these considered 'acidic soils'? I think soil scientists (which I am not) consider pH 5.5 to be the cut-off for acidic soils.

232 - Perhaps a pedantic point, but does 'inhibited' implies that the AOA were there and then stopped functioning due to salinity. Is it more a selection against AOA rather than inhibition.

283 - 'highly significant' - give a p value if describing as significant?

302 - I can't imagine a situation how they could be present in the soil if they were not active nitrifiers

304 - correlation

375 - 'homogeneous living style' - I don't know what this means. In terms of being "both acidic-living and slow-growing organisms", this seems a bit tenuous and I would consider making these links. We can assume that a large proportion (majority?) are slow growing, and the organisms in an individual soil pH are probably adapted to occur in the same range of pH (for all organisms present). Finally, I think Acidobacteria are also found in neutral pH soils and the name is a misnomer.

377 - 'supported this homogeneity'. I don't know what this means.

410 - 'we found a prevalent co-occurrence.....suggesting a niche specialization of these clades'. This doesn't make much sense to me. Are you saying they are present in the same niche?

420 - 'Blasting' is probably not a real verb in this context and slang.

423 - suggest 'linked pairs' rather than 'couples'

426 - genes

426 - contaminants

427 - limited number of published genomes

428 - suggest replacing 'need' with 'required'

464 - add rRNA

Response to editor and reviewers

We would like to thank the editor and both reviewers for the valuable remarks and comments. Your kind consideration of this manuscript for publication in **mSystems** would be greatly appreciated. We have revised this manuscript carefully according to your comments and suggestions.

In this revision, we mainly answered the questions to address the reviewers' concerns and made revisions according to the editor and reviewers' comments. All the revisions were marked in the "Marked-Up Manuscript" file. Details are shown in the following responses.

Response to editor:

Thank you for submitting your manuscript to mSystems. We have completed our review and I am pleased to inform you that, in principle, we expect to accept it for publication in mSystems. However, acceptance will not be final until you have adequately addressed the reviewer comments. I initially invited three reviewers, but the second reviewer did not provide their review in a timely fashion, so I'm making my decision based upon comments from the first and third reviewers. While both reviewers thought the manuscript was relatively well-written, they identified several grammatical and language errors throughout the text. Please ensure that you carefully revise the text to improve the quality of the writing. Another concern brought up by the third reviewer was the usefulness of the network analysis. Similar to the reviewer, I think this section should be removed from the manuscript, as it doesn't appear to add to the paper in a substantive way. I suggest you either delete that section entirely or move it to the supplement.

Response: Thanks for your positive feedbacks. We have carefully revised the text according to reviewers' comments and details can be found in the following responses to reviewers. We agree that the network doesn't appear to add much to the paper, and we therefore deleted the whole section, as well as the corresponding paragraphs in the Introduction and Methods. These changes can be seen in the "Marked-Up Manuscript" file. However, we still consider that our database provides the possibility to study the potential interactions between the *amoA* clades and the other microorganisms. We therefore added the following statements to address this potential in the revised manuscript:

"With our database, we can define the *amoA* clades by 16S rRNA genes while accessing the

information of other microbial groups using the same dataset. One potential application could be utilizing the co-occurrence network to reveal the potential interactions between *amoA* clades and the other microorganisms, thus better understanding the ecological roles of these clades in different environments.” (L190-194 in the “Marked-Up Manuscript” pdf file; L168-172 in the merged pdf file)

Response to reviewer #1:

Reviewer #1 (Comments for the Author):

*In this manuscript, the authors present an innovative approach to link the curated taxonomy of the archaeal ammonia monooxygenase gene with each corresponding 16S rRNA ASVs. Then, they used these connections to infer shifts in AOA subclade abundance with changes in environmental conditions across several peatland soil in N. Germany. Using network analysis, they also tracked changes in bacterial community structure with correlated abundance patterns to the *amoA* set of ASVs, and used trait information for these bacterial clades to propose putative reasons for their correlated abundance patterns. This is an interesting piece and enables the assessment of the degree of niche specialization of AOAs here applied to peatlands, but it will be exciting to see how these patterns hold in other environments.*

*The development of the database linking *amoA* genes to their 16S rRNA counterparts is rigorous, and the subsequent application to understanding AOA community shifts is comprehensive. Aside from several grammatical errors (see non-exhausted list below), one minor concern or consideration to potentially address is that while this database certainly further delineates AOAs into finer subclades, it may still lack some key members by virtue of the fact that these *amoA*-16S MAGs (or long contigs) are missing (lines 422-423).*

Assuming that MAGs are more often formed from more abundant clades, the database is missing rarer members. I guess this isn't problematic as the objective here is to understand function-community relationships, and these members are less critical to those patterns.

Response: Thanks for your valuable remarks. Our 16S rRNA-*amoA* pairs are not only from MAGs, but also from long sequences/contigs. Even though these sequences might still not cover many rare members, this is not problematic as suggested in your comments. On the other hand,

rare members in the peat (or any other environments) do not necessarily reflect missing 16S rRNA-*amoA* pairs in the reference database, which was constructed from published datasets of a variety of different microbiomes and ecosystems with differing AOA community composition and abundance.

However, another set of missing MAGs (or long contigs) are those for which the amoA and 16S genes are very distantly spaced, or have high homopolymer stretches between them? I am naive to the conservation of the basepairs between these two genes, but are there ITS-like regions? If so, it is conceivable that AOAs could still be missing from the database because amoA-16S MAGs don't exist yet. Again, this may not be a reasonable concern, but if it is, maybe it is worth mentioning somewhere that despite developing a comprehensive database, some key members may still be missing.

Response: Thanks for your valuable questions. In general, there is no ITS like sequences spacer between 16S rRNA and *amo* genes. Rather the genes can be located very far away from each other. However, with current binning methods in MAG construction a continuous DNA stretch between the genes is not required. Moreover, in our manuscript, we did not argue that all MAGs lack 16S rRNA genes. Our database does include many sequences extracted from MAGs as shown in Supplemental File 2, where some of the accession numbers refer to MAGs from NCBI. To mention this point, we added following statements in the revised manuscript:

“Despite developing a comprehensive database, some AOA MAGs that might represent key members are missing in the database due to the lack of the 16S rRNA gene (details in Methods), but this flaw will eventually disappear as the database will be maintained with new genomes being sequenced.” (L146-149 in the “Marked-Up Manuscript” pdf file; L126-129 in the merged pdf file)

Line-specific comments:

457 - 90% seems like a low threshold, but maybe it is sufficient?

Response: This is the default threshold used by uclust. We think this threshold is sufficient for the current size of the database, as minimally three hits are needed for an assignment. Also, we here mainly focus on the level of sub-clades (of Order), and we do not need that high resolution on the taxonomy. However, as the database might keep growing, the threshold might also need to

be adjusted. No matter how database changes, users can always adjust the threshold according to their own needs.

506-508 - *I assume the Hellinger-transform data was used as input to both Deseq and the ordination, not just the latter?*

Response: Hellinger-transformation was only used to normalize the data used for ordination analysis to get rid of the impact of uneven sample sizes. The DESeq implementation itself includes a step to normalize the data through estimating the size factors to render counts from different samples, which may have been sequenced to different depths, comparable. This statement can be found in this publication: <http://dx.doi.org/10.1186/gb-2010-11-10-r106>. To avoid the confusion, we revised this sentence in the revised manuscript (L538-540 in the "Marked-Up Manuscript" pdf file; L430-433 in the merged pdf file).

527 - *"Containmination"*

Response: We have revised this wrong spelling (L580 in the "Marked-Up Manuscript" pdf file; L469 in the merged pdf file).

Fig 3 - *"Corefficient" (I actually thought this was a new term)*

Response: We have revised this wrong spelling in Fig. 3.

Reviewer #3 (Comments for the Author):

This is a very welcome study which aims to provide unity between amoA and 16S rRNA-gene based phylogenies in molecular ecology surveys of AOA. While the phylogenies produced from these two genes have long been recognized as highly congruent, the inability to directly use, for example, the Alves et al. classification system with 16S rRNA data beyond order-level assignation has limited the interpretation. The demonstration of the using this 16S rRNA-based approach for looking at the distribution of clades in the different samples across time and depth (Figure 2) was really very nice and clear and shows that this will be a welcome tool for the community. While the manuscript is generally clear and logically structured, I do have some minor comments.

1. The term '16S-predicted *amoA* clades' is used in several places in the manuscript. While I understand what it means (i.e. clades of AOA that were named from an *amoA*-based phylogeny but are now identified from their paired/corresponding 16S rRNA sequences) it is a bit confusing and doesn't quite make sense. At this point in the manuscript it is pretty clear (to me) that these the 16S rRNA gene-defined clades correspond to the same groups (names) as the original *amoA*-based designation. Can you just say 16S rRNA gene-defined clade?

Response: Thanks for your considerable suggestion. We have changed all “16S-predicated *amoA* clades” to “16S rRNA gene-defined *amoA* clades” (Changes are marked in the “Marked-Up Manuscript” file).

2. The study makes heavy use of correlation to make predictions of which physicochemical properties influence the individual clades. I think this is okay (although terms like "Associations" (title in line 218) are a little strong for me) if being used for making hypotheses for experimentally testing in the future. However, I'm less sure the network analysis is as compelling or informative as the rest of the manuscript, with points in the discussion understandably vague. These network correlations are not proof (or necessarily an indication) of interaction, and statements that correlating Acidobacteria and Nitrosotalea populations are both slow-growing and acidophilic doesn't really tell us anything (in my opinion). I would consider simply removing the network component, as the (co-?) correlations seem to be with many different taxa and not really indicating any causal or functional interactions.

Response: Thanks for your considerable suggestion. We have removed the whole network section from the manuscript, according to both your and the editor's suggestion. Detailed reply can be found in the above response to the editor.

3. The manuscript is well written although some grammatical/structural issues occurred in the last sections of the discussion and in the M&M. For the latter I have highlighted suggested revisions for the first paragraph although these are not exhaustive.

Response: Thanks for your highlights. We have revised all points you have mentioned. We also carefully checked the whole manuscript to avoid these issues (Changes are marked in the “Marked-Up Manuscript” file).

4. *Figure 1A - what do the coloured sequence names imply? For example, in the Nitrosotaleales, the sequence names are in green. I guess this is a habitat colour-coding, but details are missing in the key or legend.*

Response: Thanks for your useful question. We realized that we missed some Greek letters in Fig. 1B which correspond to the color code of the names in Fig. 1A. We added the letters and a new sentence “The names of the sequences or ASVs are colored with bars according to sub-clades, in line with the colors of the symbols in B).” in the figure legend (L855-856 in the “Marked-Up Manuscript” pdf file; L724-725 in the merged pdf file).

Minor comments

34 - *as 'a' marker*

Response: We have revised it according to your suggestion (L34 in the “Marked-Up Manuscript” pdf file; L34 in the merged pdf file).

43, 238 *etc.* - *Water content or status rather than condition?*

Response: We think “water status” is better, and we changed this term across the whole manuscript (Changes are marked in the “Marked-Up Manuscript” file).

67 - *The microbial transformation of ammonia..... ?*

Response: We have revised it according to your suggestion (L67 in the “Marked-Up Manuscript” pdf file; L64 in the merged pdf file).

69 - *had been*

Response: We have revised it according to your suggestion (L69 in the “Marked-Up Manuscript” pdf file; L66 in the merged pdf file).

83 - *suggest removing 'right' – redundant*

Response: We have revised it according to your suggestion (L83 in the “Marked-Up Manuscript” pdf file; L80 in the merged pdf file).

133 - suggest '16S rRNA-predicted'

Response: We have changed it to “16S rRNA gene-defined”, taking into account your previous comment (L133 in the “Marked-Up Manuscript” pdf file; L116 in the merged pdf file).

196 - I realise pH 6 is technically acidic, but in the context of soils, are these considered 'acidic soils'? I think soil scientists (which I am not) consider pH 5.5 to be the cut-off for acidic soils.

Response: The pH conditions here are defined as acidic ($\text{pH} < 6.5$), neutral ($6.5 \leq \text{pH} \leq 7.5$) and alkaline ($\text{pH} > 7.5$) according to reference (15). We have added this explanation in Table 1 in the revised manuscript (L849-850 in the “Marked-Up Manuscript” pdf file; L718-719 in the merged pdf file).

232 - Perhaps a pedantic point, but does 'inhibited' implies that the AOA were there and then stopped functioning due to salinity. Is it more a selection against AOA rather than inhibition.

Response: Yes, it is more a selection, but based on the inhibition of other AOA clades rather than NT- α . To make it clearer, we changed the saying to “...inhibited the majority of AOA community...” (L248 in the “Marked-Up Manuscript” pdf file; L222 in the merged pdf file).

283 - 'highly significant' - give a p value if describing as significant?

Response: We have added the P value in the revised manuscript (L300 in the “Marked-Up Manuscript” pdf file; L271 in the merged pdf file).

302 - I can't imagine a situation how they could be present in the soil if they were not active nitrifiers

Response: Thanks for the remark. The sequences were obtained from the DNA not RNA. The presence of their DNA does not necessarily mean that they are active. However, what we want to indicate here is that since these two clades showed positive and negative correlations with nitrate and ammonium, respectively, they might actually be using the ammonium to produce nitrate and thus being active.

304 – correlation

Response: We have revised it according to your suggestion (L320 in the “Marked-Up

Manuscript" pdf file; L292 in the merged pdf file).

375 - *'homogeneous living style' - I don't know what this means. In terms of being "both acidic-living and slow-growing organisms", this seems a bit tenuous and I would consider making these links. We can assume that a large proportion (majority?) are slow growing, and the organisms in an individual soil pH are probably adapted to occur in the same range of pH (for all organisms present). Finally, I think Acidobacteria are also found in neutral pH soils and the name is a misnomer.*

Response: Thanks for your comments. We agree that not all members in Acidobacteria are acidic-living or slow-growing. However, we have removed the whole network section according to the editor's and your previous suggestion.

377 - *'supported this homogeneity'. I don't know what this means.*

Response: We have removed the whole network section according to the editor's and your previous suggestion.

410 - *'we found a prevalent co-occurrence.....suggesting a niche specialization of these clades'. This doesn't make much sense to me. Are you saying they are present in the same niche?*

Response: We have removed the whole network section according to the editor's and your previous suggestion.

420 - *'Blasting' is probably not a real verb in this context and slang.*

Response: Thanks for pointing out this mistake. We removed "...based on blasting" as we actually used uclust for the assignment as mentioned in the following sentences. (L447 in the "Marked-Up Manuscript" pdf file; L356 in the merged pdf file)

423 - *suggest 'linked pairs' rather than 'couples'*

Response: We have changed all the corresponding "couples" to "linked pairs" in the revised manuscript. (L452, 462 in the "Marked-Up Manuscript" pdf file; L360, 369 in the merged pdf file)

426 – *genes*

Response: We have revised it according to your suggestion (L454 in the “Marked-Up Manuscript” pdf file; L362 in the merged pdf file).

426 – *contaminants*

Response: We have revised it according to your suggestion (L455 in the “Marked-Up Manuscript” pdf file; L363 in the merged pdf file).

427 - *limited number of published genomes*

Response: We have revised it according to your suggestion (L455 in the “Marked-Up Manuscript” pdf file; L364 in the merged pdf file).

428 - *suggest replacing 'need' with 'required'*

Response: We have revised it according to your suggestion (L456 in the “Marked-Up Manuscript” pdf file; L365 in the merged pdf file).

464 - *add rRNA*

Response: We have revised it according to your suggestion (L494 in the “Marked-Up Manuscript” pdf file; L399 in the merged pdf file).

Other revisions:

For all the grammar and spelling mistakes mentioned by the reviewers, we have carefully checked the whole manuscript, and corrected if found. Some other minor revisions were also conducted to improve the manuscript. These changes are marked in the “Marked-Up Manuscript” file.

We have removed Fig. S1 as it was already published in the mentioned references. We also changed the names and order of the other supplemental material files according to the journal’s guidelines.

We uploaded the new versions of Supplemental File 1 and 2 (now Data Set S1 and S2), as we realized that the old files are not the correct (up-to-date) ones.

August 5, 2021

Dr. Haitao Wang
University of Greifswald
Greifswald
Germany

Re: mSystems00546-21R1 (Linking 16S rRNA gene classification to amoA gene taxonomy reveals environmental distribution of ammonia-oxidizing archaea clades in peatland soils)

Dear Dr. Haitao Wang:

Thank you for submitting your manuscript to mSystems. We have completed our review and I am pleased to inform you that, in principle, we expect to accept it for publication in mSystems. However, acceptance will not be final until you have adequately addressed a few remaining minor comments from Reviewer 3 (see below).

Preparing Revision Guidelines

For complete guidelines on revision requirements for your article type, please see the journal Article Types requirement at <https://journals.asm.org/journal/mSystems/article-types>. **Submissions of a paper that does not conform to mSystems guidelines will delay acceptance of your manuscript.**

Sincerely,

Sean Gibbons

Editor, mSystems

Journals Department
Reviewer comments:

Reviewer #1 (Comments for the Author):

In the revised manuscript, the authors have carefully and thoroughly addressed the feedback from my review and the other reviewer's recommendations, as well as responded in kind to the clarifications regarding MAGs.

It is a well written manuscript of a high quality study, and the resultant database will certainly be useful for others.

Reviewer #3 (Comments for the Author):

Not a major point, but in my response to my previous suggestion that '16S rRNA gene-defined clade' is used instead of '16S-predicated amoA clades', this has been revised to '16S rRNA gene-defined amoA clades'. I suspect this is me not being clear, for which I apologise, but this change doesn't make much difference. It is fine as it is, but my point was that, when discussing for example, NS- γ or NS- δ clades, with your new approach, you can either identify these clades either from an amoA gene sequence, or from a 16S rRNA gene sequence. When NS- δ is predicted from a 16S rRNA gene sequence, it seems more logical to me to say that it was a '16S rRNA gene-detected clade', without including the word amoA to highlight that the clade was first defined from amoA genes.

222 - 'High salinity inhibited the majority of AOA community' I don't mean to labour the point, but I don't think inhibition is very accurate. I still think this is more accurate to say something like high salinity selected for NT- α representatives adapted to high salt concentrations, or other clades did not possess adaptations for growth in saline conditions. For example, you could say throughout this manuscript that the absence of particular clades in any other samples was the result of other soil properties inhibiting them.

269 - 'Interestingly, the correlation profile of the clade NS-UD was more similar to the NP clades

profiles rather than the NS clades. The lack of a significant correlation to ammonium might indicate that it is not among the active nitrifiers in these sites.' I paste below my previous query and response below. While I understand that these data are from DNA analyses, which do not indicate in situ activity or rates, my point is that while you find no correlation with ammonium or nitrate concentrations, how else would you explain the presence of a population of ammonia oxidisers in a soil if they do not oxidise ammonia when they are present i.e. they wouldn't be there if they were not active under some conditions. If you are proposing that they may have another mode of energy metabolism explaining their presence, that is another matter, but if you consider them to be bona fide ammonia oxidisers, then their presence can only be explained by being active, at some point in time, in that soil, or they would simply not be there.

[Previous question and response: 302 - I can't imagine a situation how they could be present in the soil if they were not active nitrifiers

Response: Thanks for the remark. The sequences were obtained from the DNA not RNA. The presence of their DNA does not necessarily mean that they are active. However, what we want to indicate here is that since these two clades showed positive and negative correlations with nitrate and ammonium, respectively, they might actually be using the ammonium to produce nitrate and thus being active.]

364 - a limited number

364 - pairs

267 - the most closely

416 - the Silva

Response to reviewers

We would like to thank both reviewers again for their positive feedbacks. We have revised this manuscript according to the several comments from Reviewer 3. All the revisions are marked in the "Marked-Up Manuscript" file. Details are shown in the following responses.

Reviewer comments:

Reviewer #1 (Comments for the Author):

In the revised manuscript, the authors have carefully and thoroughly addressed the feedback from my review and the other reviewer's recommendations, as well as responded in kind to the clarifications regarding MAGs.

It is a well written manuscript of a high quality study, and the resultant database will certainly be useful for others.

Reviewer #3 (Comments for the Author):

Not a major point, but in my response to my previous suggestion that '16S rRNA gene-defined clade' is used instead of '16S-predicated amoA clades', this has been revised to '16S rRNA gene-defined amoA clades'. I suspect this is me not being clear, for which I apologise, but this change doesn't make much difference. It is fine as it is, but my point was that, when discussing for example, NS- γ or NS- δ clades, with your new approach, you can either identify these clades either from an amoA gene sequence, or from a 16S rRNA gene sequence. When NS- δ is predicted from a 16S rRNA gene sequence, it seems more logical to me to say that it was a '16S rRNA gene-detected clade', without including the word amoA to highlight that the clade was first defined from amoA genes.

Response: Thanks for explaining in more details. We have revised these phrases accordingly. They are called either "16S rRNA gene-defined clade(s)" or "*amoA* clade(s)" in the revised

manuscript. Changes can be found in the in the “Marked-Up Manuscript” pdf file.

222 - 'High salinity inhibited the majority of AOA community' I don't mean to labour the point, but I don't think inhibition is very accurate. I still think this is more accurate to say something like high salinity selected for NT- α representatives adapted to high salt concentrations, or other clades did not possess adaptations for growth in saline conditions. For example, you could say throughout this manuscript that the absence of particular clades in any other samples was the result of other soil properties inhibiting them.

Response: Thanks for the suggestion. We have changed the saying as following:

“High salinity **showed a selection** for the AOA community, as shown by the fact that few clades but NT- α were present in the coastal fen (Fig. 2). The presence of NT- α indicated their **adaption** to saline conditions, ...” (L226-228 in the “Marked-Up Manuscript” pdf file; L225-227 in the merged pdf file).

269 - 'Interestingly, the correlation profile of the clade NS-UD was more similar to the NP clades profiles rather than the NS clades. The lack of a significant correlation to ammonium might indicate that it is not among the active nitrifiers in these sites.' I paste below my previous query and response below. While I understand that these data are from DNA analyses, which do not indicate in situ activity or rates, my point is that while you find no correlation with ammonium or nitrate concentrations, how else would you explain the presence of a population of ammonia oxidisers in a soil if they do not oxidise ammonia when they are present i.e. they wouldn't be there if they were not active under some conditions. If you are proposing that they may have another mode of energy metabolism explaining their presence, that is another matter, but if you consider them to be bona fide ammonia oxidisers, then their presence can only be explained by being active, at some point in time, in that soil, or they would simply not be there.

[Previous question and response: 302 - I can't imagine a situation how they could be present in the soil if they were not active nitrifiers

Response: Thanks for the remark. The sequences were obtained from the DNA not RNA. The presence of their DNA does not necessarily mean that they are active. However, what we want to indicate here is that since these two clades showed positive and negative correlations with nitrate and ammonium, respectively, they might actually be using the ammonium to produce nitrate and

thus being active.]

Response: We agree that their presence could be explained as being active at some time point in that soil, but probably not all the time (which needs to be identified with the RNA data). We also think that it is tricky to say “being active or not” based on the DNA data, regardless of the correlations. Therefore, we have removed these two interpretations in the revised manuscript. (L276-277 and L296 in the “Marked-Up Manuscript” pdf file).

364 - a limited number

Response: We have revised it according to your suggestion (L370 in the “Marked-Up Manuscript” pdf file; L367 in the merged pdf file).

364 – pairs

Response: We have revised it according to your suggestion (L371 in the “Marked-Up Manuscript” pdf file; L368 in the merged pdf file).

267 - the most closely

Response: We believe that this comment is related to L367, and we have revised it according to your suggestion (L374 in the “Marked-Up Manuscript” pdf file; L371 in the merged pdf file).

416 - the Silva

Response: We have revised it according to your suggestion (L423 in the “Marked-Up Manuscript” pdf file; L420 in the merged pdf file).

August 12, 2021

Dr. Haitao Wang
University of Greifswald
Greifswald
Germany

Re: mSystems00546-21R2 (Linking 16S rRNA gene classification to amoA gene taxonomy reveals environmental distribution of ammonia-oxidizing archaea clades in peatland soils)

Dear Dr. Haitao Wang:

Your manuscript has been accepted, and I am forwarding it to the ASM Journals Department for publication. For your reference, ASM Journals' address is given below. Before it can be scheduled for publication, your manuscript will be checked by the mSystems senior production editor, Ellie Ghatineh, to make sure that all elements meet the technical requirements for publication. She will contact you if anything needs to be revised before copyediting and production can begin. Otherwise, you will be notified when your proofs are ready to be viewed.

As an open-access publication, mSystems receives no financial support from paid subscriptions and depends on authors' prompt payment of publication fees as soon as their articles are accepted. =

Publication Fees:

We recognize that the video files can become quite large, and so to avoid quality loss ASM suggests sending the video file via <https://www.wetransfer.com/>. When you have a final version of the video and the still ready to share, please send it to Ellie Ghatineh at eghatineh@asmusa.org.

Sincerely,

Sean Gibbons
Editor, mSystems

Journals Department
Data Set S4: Accept
Data Set S1: Accept
Table S2: Accept
Data Set S2: Accept
Fig. S2: Accept
Table S1: Accept
Data Set S3: Accept
Fig. S3: Accept
Fig. S1: Accept
Fig. S4: Accept